# The Theory of Surface-Enhanced Raman Spectroscopy on Organic Semiconductors: Graphene

**DOI:** 10.3390/nano12162737

**Published:** 2022-08-09

**Authors:** John R. Lombardi

**Affiliations:** Department of Chemistry and Biochemistry, The City College of New York, New York, NY 10031, USA; lombardi@sci.ccny.cuny.edu

**Keywords:** SERS, organic semiconductors, graphene, Raman enhancement, density of states

## Abstract

Drawing on a theoretical expression previously derived for general semiconductor substrates, we examine the surface-enhancement of the Raman signal (SERS) when the substrate is chosen to be monolayer graphene. The underlying theory involves vibronic coupling, originally proposed by Herzberg and Teller. Vibronic coupling of the allowed molecular transitions with the charge-transfer transitions between the molecule and the substrate has been shown to be responsible for the SERS enhancement in semiconductor substrates. We then examine such an expression for the Raman enhancement in monolayer graphene, which is dependent on the square of the derivative of the density of states of the graphene. On integration, we find that the discontinuity of the density-of-states function leads to a singularity in the SERS intensity. Knowledge of the location of this resonance allows us to maximize the Raman intensity by careful alignment of the doping level of the graphene substrate with the charge-transfer transition.

## 1. Introduction

It has been observed that when a molecule is oriented close to the surface of a metallic nanoparticle such as Ag and Au, the intensity of the Raman spectrum (which is normally quite weak) is often enhanced by many orders of magnitude [1]. This area of research was termed surface-enhanced Raman scattering (SERS). Routinely the enhancement factors were determined to be on the order of 10^6^, but with subsequent investigations, techniques were explored which reached factors of up to 10^12^. These remarkable enhancements were sufficiently sensitive to enable detection of a single molecule [2]. Such observations spurred the adoption of Raman spectroscopy as a highly sensitive tool for the detection and identification of molecules in minute quantities. The source of such enhancements has been shown to involve coupling of several resonances in the molecule–metal system [3]. These included a surface plasmon resonance located in the metal nanoparticle, molecular resonances centered in the adsorbed molecule, and new resonances involving charge-transfer between levels of the molecule and the metal Fermi level [4]. Such charge-transfer resonances were found to be enhanced by intensity borrowing from the molecular resonances by means of Herzberg–Teller vibronic coupling. This latter involves a breakdown of the Born–Oppenheimer approximation.

After several years, the search for new SERS substrates led to the exploration of inorganic semiconductors [5,6] and subsequently organic semiconductor [7,8] substrates as well. However, since semiconductors usually do not have plasmon resonances near the visible region of the spectrum, the observed enhancement factors were considerably lower than those observed with metallic substrates. These were typically on the order of 10^3^ (although, a few exceptions have been observed [9]), and only rarely much above 10^5^. This drawback was often considered worth accepting since semiconductor substrates tend to be much more stable than nanoscale metal substrates, and they were also much easier to reproducibly synthesize. Furthermore, the semiconductor substrates could often be fabricated in such a way as to optimize the location of the charge-transfer contribution to the overall enhancement. These advantages often outweigh the disadvantages of lower enhancement. The theoretical analysis was marshalled to find ways to optimize the SERS signal by taking advantage of the Herzberg–Teller coupling mechanism [10].

In this work, we will apply the theory of SERS to the adoption of graphene substrates. Graphene has many special properties, which encourage widespread use for many applications. Numerous attempts have been made to take advantage of these properties [11,12,13,14,15]. However, none of these explorations purposefully took advantage of the theoretical foundations of SERS to optimize the signal. It is the objective of this article to apply the theory to develop a route to optimize the Raman enhancements for molecules adsorbed on graphene substrates.

## 2. Optimum SERS Intensities from a Molecule Adsorbed on Mono-Layer Graphene

More than 50 years ago, Albrecht [16] derived a general expression for Raman intensities, by which the polarizability of the system (*α*) could be written:(1)α=A+B+C
where the *A* term involves Franck–Condon sums, and the *B*, *C* terms represent the inclusion of Herzberg–Teller coupling. Since then, this formalism has been extended to surface-enhanced Raman spectroscopy [3], in which the molecular (*m*) and charge-transfer (*CT*) optical transitions are coupled by a Herzberg–Teller coupling constant (*h_m-CT_*). Whenever the *CT* transitions involved are from a level in the molecule to a band in the semiconductor (in this case the *B*-term), we may replace the sum over all states with an integral over the semiconductor levels such that:(2)$$B\left(\omega \right)=-\left(\frac{2}{\hbar^{2}}\right)\frac{\mu_{m}\mu_{CT}h_{m-CT}\langle i|Q|f\rangle }{\gamma_{CT}^{2}}\int_{-\infty }^{\infty }\frac{\omega_{CT}\omega_{m}+\omega^{2}}{\left(\omega_{m}^{2}-\omega_{CT}^{2}\right)}\rho \left(\omega_{CT}\right)d\omega_{CT}\;=-B_{0}\int_{-\infty }^{\infty }\frac{\omega_{CT}\omega_{m}+\omega^{2}}{\left(\omega_{m}^{2}-\omega_{CT}^{2}\right)}\rho \left(\omega_{CT}\right)d\omega_{CT}$$

In this formulation:

$$B_{0}=\left(\frac{2}{\hbar^{2}}\right)\frac{\mu_{m}\mu_{CT}h_{m-CT}\langle i|Q|f\rangle }{\gamma_{CT}^{2}}$$
and *μ_m_*, *μ_CT_*, and *h_m-CT_* represent the electronic transition moment of the molecule (*μ_m_*), the charge-transfer transition moment (*μ_CT_*), and the Herzberg–Teller coupling constant (*h_m-CT_*), respectively. *Q* is the displacement of any chosen molecular normal mode between the ground state vibrational levels (*i*,*f*), and *γ_CT_* is the band-width of the charge-transfer transition [17]. Here *ρ* is the density of semiconductor levels. Integrating by parts (assuming *ρ* vanishes at ±∞) gives:(3)$$B=B_{0}\int_{-\infty }^{\infty }\frac{\omega_{CT}\omega_{m}+\omega^{2}}{\left(\omega_{m}^{2}-\omega_{CT}^{2}\right)}\frac{\partial \rho }{\partial \omega_{CT}}d\omega_{CT}$$

The density-of-states function for graphene in the vicinity of the graphene Dirac point [18,19] (Figure 1) is:(4)$$\rho \left(\omega_{CT}\right)=\frac{2A_{c}}{\pi v_{F}^{2}}\left|\omega_{CT}\right|=b\;\left|\omega_{CT}\right|=\;b\omega_{CT}H\left(\omega_{CT}\right)\;\;\;$$
where we have taken $b=2A_{c}/\pi v_{F}^{2}$ and *A_c_* is the area of a unit cell of graphene, *v_F_* is the Fermi velocity, and *H* is the Heaviside (step) function. For convenience, *H* may be chosen symmetrically such that *H*(*x*) = +1 for x > 0, and −1 for x < 0.

We use the fact that the derivative of the Heaviside function is simply the Dirac *δ* ($\omega_{CT}$) function, leading to:(5)$$\frac{\partial \rho }{\partial \omega_{CT}}=\;b\;\omega_{CT}\delta \left(\omega_{CT}\right)+bH\left(\omega_{CT}\right)=bH\left(\omega_{CT}\right)$$
(6)$$B=-B_{0}b\int_{-\infty }^{\infty }\frac{\omega_{CT}\omega_{m}+\omega_{CT}^{2}}{\left(\omega_{m}^{2}-\omega_{CT}^{2}\right)}H\left(\omega_{CT}\right)d\omega_{CT}=-B_{0}b\int_{0}^{\omega_{f}}\frac{\omega_{CT}}{\left(\omega_{m}-\omega_{CT}\right)}d\omega_{CT}$$

It is common for the Dirac point (Figure 2) to be chosen as the origin. The integral (6) then spans the limits (0, $\omega_{f}$) where $\hbar \omega_{f}$ is the top edge of the filled electronic levels.

We may call this the charge transfer (*CT*) band. Evaluating the integral (6), we obtain:(7)$$B\left(\omega_{f}\right)=-bB_{0}\omega_{m}ln\left|\frac{\omega_{f}-\omega_{m}+i\gamma_{CT}}{\omega_{f}}\right|+\omega_{f}.$$

Note that the discontinuity in the density-of-states function in graphene at the Dirac point leads to a singularity in the SERS intensity. This type of logarithmic resonance has previously been anticipated by the observation of logarithmic singularities in metallic SERS (ref. [4]), upon the introduction of a band-edge step function in the density of states. These functions gave excellent fits to the experimental metal-SERS excitation profiles. Of course, the excited state of the *CT* transition is of finite lifetime, and for practical calculations we must introduce a finite linewidth ($\gamma_{CT}$) to the resonance. Importantly, we have identified the location of the optimum SERS signal, such that the charge-transfer transition in the conduction band terminates at the highest filled level of graphene ($\omega_{m}=\omega_{f}$). This can be achieved, for example, by adjusting the doping level of the graphene sheet, and setting the laser such that $\omega_{CT}=\omega_{f}=\omega_{m}$.

This is illustrated in Figure 2, where we show an energy level diagram of the graphene–molecule system. The term *ω_m_* (the molecular resonance) is represented by an arrow connecting an occupied level of the ground state (I) to an excited state (K) transition in the molecule. The Herzberg–Teller effect involves vibronic coupling between this (presumably allowed) molecular transition and one of the two possible charge-transfer transitions *ω_CT_*. Transitions either from the ground state of the molecule to one of the empty levels of graphene (B-term) or from the highest filled level of graphene to an unfilled excited level of the molecule (C-term) are represented by *ω_CT_*. Only the former is shown in Figure 2. The enhancement of the Raman signal is then obtained by the so-called “borrowing” of intensity from the allowed molecular transition by the (normally weak) charge-transfer transition [8]. We can therefore optimize the SERS signal by setting the Raman excitation laser at (or near) the charge-transfer transition (*ω_CT_*). A similar result is found for the analogous derivation for the C term, which we will not repeat.

As an example, let us take the molecule copper phthalocyanine (CuPc) on a monolayer of graphene. This system is illustrated in Figure 3, where we show an energy level diagram. CuPc has an ionization potential of 6.38 eV, so we take the highest occupied molecular orbital (HOMO) at −1.88 eV wrt the vacuum level [20]. The first optically allowed transition in CuPc is located at 1.99 eV [21], so that the LUMO is located at +0.11 eV [22]. The optimum SERS excitation would then be expected at or near 1.99 eV (623 nm), and the (*n*-) doping level of the graphene monolayer should therefore be set at −0.11 eV. This illustrates the value of obtaining good excitation profiles of molecules of interest on graphene as a function of doping level.

## 3. Discussion

We have now illustrated the value of knowing the density-of-states function of an organic semiconductor. The SERS intensity depends crucially on the derivative of the density of states as a function of energy, and consequently, wherever this *derivative* becomes large. Near a singularity in this function, we predict a very large increase. Such a singularity occurs due to a discontinuity in the density-of-states function itself. In monolayer graphene there is such a discontinuity near the Dirac point. In Figure 2, we clearly see this discontinuity at E_F_ = 0. Such considerations point us in the direction of predicting the conditions needed to optimize the SERS intensity. We can often obtain information concerning the density-of-states function or its derivative by measuring the appropriate excitation profiles.

There are several examinations of density-of-states functions using the molecule CuPc (copper phthalocyanide). One study [19] utilized the four wavelengths 488, 514.5, 633, and 785 nm to derive a crude excitation profile in Au-graphene systems. Using a Gaussian line-shape best fit of the intensities against the excitation wavelength, an intensity optimum at 1.99 eV (623 nm) was obtained. This also shows that the density of states is not significantly perturbed by adsorption of CuPc. A more sophisticated experiment, in which a continuous wave laser (OPO) was utilized to examine the SERS output as a function of excitation wavelength has been carried out with CuPc on a gold surface [20]. The excitation profile was shown to depend on the symmetry of the Raman lines observed. It was found that totally symmetric modes (a_1g_) displayed a maximum at 475 nm (2.61 eV), while the non-totally symmetric modes (b_2g_) peaked at 605 nm (2.05 eV). Ling et al. [23,24] have obtained extensive excitation profiles on CuPc with monolayer graphene and observed peaks at 2.00 eV. However, it can be seen from the previous section that for graphene substrates a more useful excitation profile is obtained as a function of the doping level [25].

## Figures and Tables

**Figure 1 nanomaterials-12-02737-f001:**
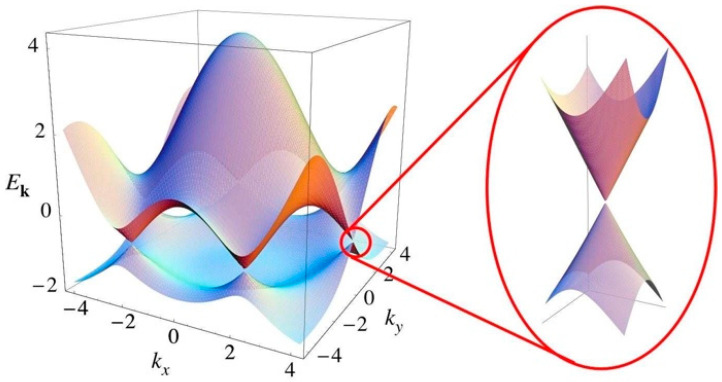
Energy spectrum of monolayer graphene. Right: a zoomed in view of the energy bands close to the Dirac point.

**Figure 2 nanomaterials-12-02737-f002:**
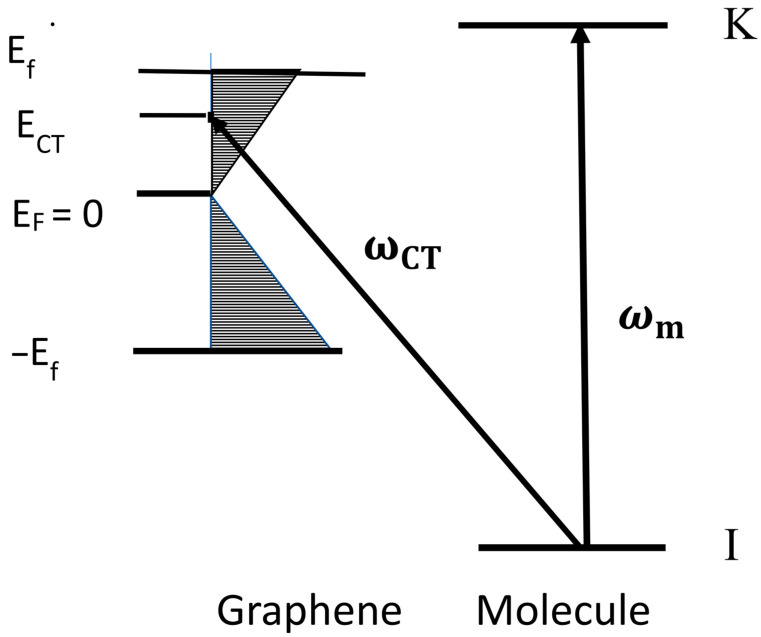
Energy level system of a molecule on the surface of monolayer graphene. I and K are the HOMO and LUMO of the molecule (more generally, they can represent any pair of filled and unfilled level in the molecule). On the left are various bands of monolayer graphene. *E_F_* is the Fermi level, usually taken to be the origin of the energy scale. Then, for graphene, *E_F_* = 0.0 eV. E_f_ is the lowest filled or highest unfilled level of graphene. *E_CT_* is the energy of charge transfer (such that $\hbar \omega_{CT}=E_{CT}-E_{I}$) for transfer from the molecule to the substrate (or $\hbar \omega_{CT}=E_{K}-E_{\mathrm{CT}}$ for transfer from the substrate to the molecule). *E_f_* is the highest filled or lowest unfilled level of graphene.

**Figure 3 nanomaterials-12-02737-f003:**
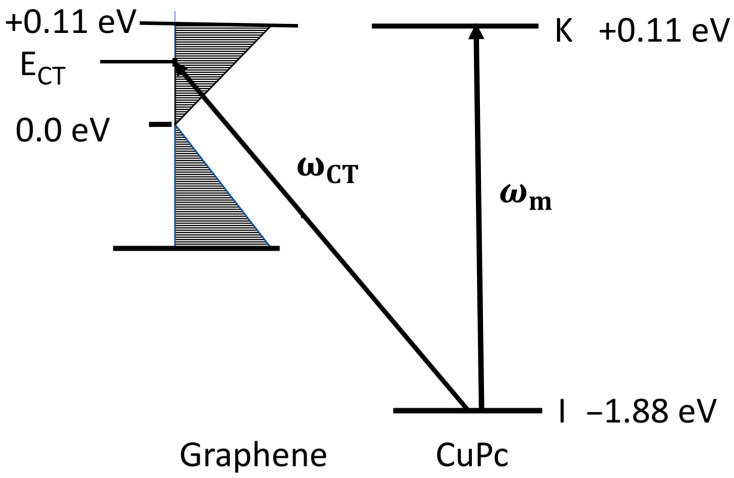
Optimization of the SERS intensity. Energy level system of CuPc on the surface of doped monolayer graphene. *I* and *K* are the HOMO and LUMO of the molecule (more generally, they can represent any filled or unfilled orbitals in the molecule). On the left are various bands of monolayer graphene. *E_F_* is the Fermi level, usually taken to be the origin of the energy scale such that usually *E_F_* is taken to be zero. *E_CT_* is the energy of charge transfer transition (such that $\hbar \omega_{CT}=E_{CT}-E_{I}$) for charge transfer from the molecule to the substrate (or $\hbar \omega_{CT}=E_{K}-E_{\mathrm{CT}}$ for charge transfer from the substrate to the molecule). Then the (*n*)-doped highest filled level (band edge) is chosen to be 0.11 eV above the Fermi level to optimize the SERS signal such that $\hbar \omega_{CT}=\hbar \omega_{m}=E_{K}-E_{I}$.

## Data Availability

Not applicable.

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
