# Peer review of "The Theory of Surface-Enhanced Raman Spectroscopy on Organic Semiconductors: Graphene"

_nanomaterials, 2022, doi:10.3390/nano12162737_

Round 1

Reviewer 1 Report

This manuscript aimed to explain the SERS enhancement observed with graphene materials, by employing the theoretical expression previously derived for general semiconductor substrates. Monolayer graphene is taken as the model substrate, and the Raman enhancement is found to depend on the square of the derivative of the density of states of the graphene. The findings are constructive to the theory of SERS concerning about charge-transfer contributions, also valuable to the design of graphene-based substrates. Here are some suggestions to the authors.

(1) Various non-metal materials have been documented with SERS activities even comparable to metals by utilizing the charge-transfer resonances, such as inorganic semiconductors, graphene, conjugated compounds, metal-organic frameworks, etc. Such information is suggested to be included in the introduction part, by resorting to a recent Review focusing on the charge-transfer induced SERS (The Innovation 2020, 1, 100051).   

(2) Whether the density of state for graphene would be modified, as it is loaded with CuPc molecules?

(3) Page 4, Line 140, why Au-graphene systems are utilized here for the excitation profile examination?

(4) The excitation profile is shown to depend on the symmetry of the Raman lines. It is interesting, while more discussions are suggested for a clear understanding about such phenomenon.

Author Response

Reply to referees:

1) I have added a reference and modified the text slightly to accommodate the referees suggestion. See revised manuscript.

2) Due to geometrical factors there is only weak interaction between substrate and adsorbate, so the density of states function will not be strongly modified. A comment is added to the text.

3) This is because the Au-graphene system is the only one available.

4) See the theory of SERS article in reference 3, for insights.

Reviewer 2 Report

The manuscript by John et al. reports a study about the Raman enhancement in monolayer graphene. It demonstrates its density depends on the square of the derivative of the density of states of the graphene. The topic is exciting, and the results are promising. Therefore, after addressing the following questions, I recommend the work published in Nanomaterials.

Please provide a clear conclusion of the current study to show what we got in this study rather than listing others’ results.

Author Response

I would like to suggest that I not be asked to add a summary of the article since it is so short that any summary would be superfluous, and therefore repetitive.